# A BENCHMARK FOR VERICODING:
# FORMALLY VERIFIED PROGRAM SYNTHESIS

## ABSTRACT

We present and test the largest benchmark for *vericoding*, LLM-generation of formally verified code from formal specifications — in contrast to *vibe coding*, which generates potentially buggy code from a natural language description. Our benchmark contains 12,504 formal specifications, with 3,029 in Dafny, 2,334 in Verus/Rust and 7,141 in Lean. Of these, 6,174 are new unseen problems. We find vericoding success rates of 27% in Lean, 44% in Verus/Rust and 82% in Dafny using off-the-shelf LLMs. Adding natural-language descriptions does not significantly improve performance. We also find that LLM progress has improved progress on pure Dafny verification from 68% to 96% over the past year.

## 1 INTRODUCTION

Rapid AI progress has popularized *vibe coding*, which generates computer programs from natural language descriptions. For example, Google has reported that over 30% of its software is created this way (Google Earnings Call). Unfortunately, the resulting code can be buggy, and traditional bug hunting with test cases can typically only demonstrate the presence and not the absence of bugs, since there are too many test cases to try them all. For example, major code-testing efforts failed to prevent bugs causing an Ariane-V rocket explosion (Ariane 5 Failure) and an embarrassing security vulnerability in the Bash shell (Shellshock Bug) that was built into the Unix operating system for 25 years before being discovered. The 2024 CrowdStrike outage disrupted 8.5 million devices globally, harming airlines, hospitals, banking, broadcasting, emergency services (CrowdStrike Outage).

Fortunately, rigorous correctness guarantees can be created via *formal verification*, by generating a machine-checkable proof that code meets its human-written specifications. Unfortunately, despite a venerable history dating back to Turing (1950), formal verification remains niche, applied to only a tiny fraction of all software because it requires much more human labor than programming does.

This makes it timely to test whether AI can help, either by verifying existing code or by writing new formally verifiable code from scratch based on its specification, which we term *vericoding*. The premise of this paper is that AI will soon be able to greatly facilitate both, dramatically reducing the cost of creating bug-free software. It is easy to imagine formal verification being simply a built-in final step of future compilers, which discover code problems and attempt to fix them automatically. One can also imagine a future where humans do not need to write programs, only specs.

This optimistic premise is based on the close analogy with automated theorem proving, where AI produces formal proofs not about code but about mathematical theorems. Fueled by the advent of benchmarks totaling over 100,000 theorems, AI tools have during the last few years improved their success rate from 21% to over 82% on the MetaMath benchmark (Polu & Sutskever, 2020; Lample et al., 2022). In August 2025, Seed-Prover achieved over 50% on PutnamBench (Chen et al., 2025), proved 78.1% of formalized past mathematics olympiad problems, and scored 99.6% on the MiniF2F benchmark (up from a 50% SOTA mid 2024). Together with the Seed-Geometry engine, the models solved 5 of 6 problems at IMO 2025.

Unfortunately, formal verification sorely lacks correspondingly large benchmarks: the largest of their kind contain fewer than $10^3$ examples. There is room for expanding not only their size and diversity, but also their level of difficulty: Many examples are limited to single-function programs, and sometimes the formal specification for a program directly repeats an implementation of the algorithm. To support automation of formal verification and vericoding, the goal of the present paper

is to provide such a benchmark expansion, by assembling and testing a suite of formal specifications for Lean (Moura & Ullrich, 2021), Rust/Verus (Lattuada et al., 2024) and Dafny (Leino, 2010).

The rest of this paper is organized as follows. We summarize related work in Section 2, and describe our benchmark construction in Section 3, outlining how vericoding sources were assembled or translated from natural language documentation, vibe coding datasets and verification benchmarks. In Section 4, we quantify the ability of current LLMs to solve vericoding tasks. Special attention is given to Lean tasks, because of recent successes in AI-assisted theorem proving. For example, we explore specifications expressed as Hoare triples using a new *mvcgen* feature. We summarize our conclusions in Section 5 and provide further technical details of our translation and vericoding approaches in the Appendix in the supplementary material.

## 2 RELATED WORK

Over the past two years, there has been increased interest in constructing new benchmarks for verification (proofs from formal specifications and implementations) and vericoding (implementations and proofs from formal specifications), as seen in Table 1, much work remains. There is significant variation in the types of verification and vibe coding tasks. VerifyThisBench (Deng et al., 2025), for example, generates specs, implementations and proofs jointly from natural language descriptions. Meanwhile, VeriBench (Miranda et al., 2025) takes Python code and documentation, and generates implementations, specs and unit tests in Lean to be proved by the LLM. A novel form of vibe coding comes from the FVAPPS (Dougherty & Mehta, 2025) benchmark which contains formal specs generated from natural language descriptions. For code synthesis, the LLM is given the descriptions and the formal specs, and specs and unit tests are employed to provide some formal correctness guarantees. Meanwhile, the field of AI control (Greenblatt et al., 2024) checks AI-produced code for correctness and safety using techniques such as oversight by weaker LLMs, but this does not produce formal guarantees.

In vericoding, no natural language descriptions are provided to the LLM for code generation. In benchmarks such as CLEVER (Thakur et al., 2025) and VERINA (Ye et al., 2025), the tasks include formal spec generation from documentation, in addition to formal and proof generation. Both benchmarks also require the Lean implementation to be synthesized *before* constructing a proof of its correctness. We acknowledge that spec generation is an important problem, but focus on the task of generating implementations and formal proofs in this work. We also let the LLM generate the implementation and the proof jointly — over several iterations, the model is allowed to change the implementation to make the proof easier or correct mistakes.

The challenge of constructing large coding benchmarks lies in gathering a large base of problems, formatting them and checking them for quality. DafnyBench (Loughridge et al., 2025) builds this base from existing benchmarks such as Clover and DafnySynthesis, and from GitHub scrapes. Large language models (LLMs) and other LLMs have facilitated this. FVAPPS (Dougherty & Mehta, 2025) uses LLMs to translate Python tasks from the APPS benchmark to Lean, and to format the translations. AlphaVerus (Aggarwal et al., 2024) goes further with a self-improving framework that iteratively translates programs from Dafny to Verus and leverages feedback from the verifier. In our work, we instead use LLMs out of the box, using them not just for translation, but also for critiquing the translations and for fixing errors in them.

## 3 BENCHMARK CONSTRUCTION

We are primarily interested in constructing a benchmark for two kinds of provers: automated theorem provers (ATPs) such as Dafny and Verus, which use SMT solvers to automatically discharge verification conditions, and interactive theorem provers (ITPs) such as Lean, which use tactics to build proofs. We begin by curating some original sources, such as HumanEval, Clever, Verina, APPS, and Numpy documentation. The original sources are then translated into other languages. Lastly, the translations are compiled, parsed into different sections, and quality-checked. We include tasks with specs that are incomplete, inconsistent, or non-compilable, because spec repair is an essential part of the formal verification workflow. Further details and scripts used in our construction can be found in the Appendix and in the supplementary material.

Table 1: Recent benchmarks for theorem proving, verification, vibe coding and vericoding. Verification and vericoding benchmarks are two orders of magnitude smaller than those for theorem proving and vibe coding. We list only benchmarks for Dafny, Verus and Lean. Notable benchmarks in other languages are SV-COMP (Beyer & Strejček, 2025) and SyGuS (Alur et al., 2018) in C and Java, and FVELER (Lin et al., 2024) and AFP (Archive of Formal Proofs) in Isabelle. In comparison, we have 12504 tasks of which 6174 are new or translated from other benchmarks.

| Task | Benchmark | Language | Size |
|------|-----------|----------|------|
| Thm proving | PutnamBench (Tsoukalas et al., 2024) | Lean, Isabelle, Coq | 1709 |
| Thm proving | FormalMATH (Yu et al., 2025) | Lean | 5560 |
| Thm proving | LeanDojo (Yang et al., 2023) | Lean | 98734 |
| Thm proving | LISA (Jiang et al., 2021) | Isabelle | 183000 |
| Verification | VeriBench (Miranda et al., 2025) | Lean | 113 |
| Verification | Verus-Bench (Yang et al., 2025) | Verus | 150 |
| Verification | Verified Cogen (JetBrains-Research, 2025) | Verus (among others) | 223 |
| Verification | VerifyThisBench (Deng et al., 2025) | Dafny, Why3, etc. | 481 |
| Verification | DafnyBench (Loughridge et al., 2025) | Dafny | 782 |
| Vibe coding | HumanEval (Chen et al., 2021) | Python | 163 |
| Vibe coding | FVAPPS (Dougherty & Mehta, 2025) | Lean | 4715 |
| Vibe coding | APPS (Hendrycks et al., 2021) | Python | 10000 |
| Vibe coding | CodeContests (Li et al., 2022) | Mixed | 13610 |
| Vibe coding | HumanEval-XL (Peng et al., 2024) | Mixed | 22080 |
| Vericoding | CLEVER (Thakur et al., 2025) | Lean | 161 |
| Vericoding | VERINA (Ye et al., 2025) | Lean | 189 |

Table 2: Number of tasks for each language and source. Originals are in **bold**, and translations are not. The $*$ indicates new tasks. $X : Y$ indicates that $X$ tasks were generated during translation, and $Y$ tasks remained after compiling, formatting and quality checks. We use only the latter tasks for our experiments. We release also the problematic tasks in case there is interest to use them for other purposes, such as spec repair. The task IDs are of the form `XYdddd` where X indicates the language (Dafny, Lean, Verus), `Y` refers to the source (see the Ref column in the table) and `dddd` is a four-digit zero-padded number starting from `0000`.

| Source | Ref | Dafny | Verus | Lean | Total |
|--------|-----|-------|-------|------|-------|
| APPS (Test) | A | **883 : 677** * | 677 : 536 * | 677 : 676 * | 2237 : 1889 |
| DafnyBench | D | **929 : 443** | 442 : 440 * | 440 : 440 * | 1811 : 1323 |
| NumpyTriple | T | 603 : 603 * | 603 : 581 * | **666 : 603** * | 1872 : 1787 |
| VerifiedCogen | J | 172 : 172 * | **172 : 172** | 172 : 172 * | 516 : 516 |
| Verina | V | 157 : 157 * | 157 : 156 * | **189 : 189** | 503 : 502 |
| Bignum | B | **62 : 62** * | 62 : 62 * | 62 : 62 * | 186 : 186 |
| NumpySimple | S | 59 : 58 * | 59 : 58 * | **59 : 59** * | 177 : 175 |
| HumanEval | H | **164 : 162** | 162 : 161 * | **161 : 161**[1] | 487 : 484 |
| FVAPPS | F | | | **4715 : 4006** | 4715 : 4006 |
| **Total** | | 3029 : 2334 | 2334 : 2166 | 7141 : 6368 | **12504 : 10868** |
| | | 1936 : 1729 * | 2162 : 1994 * | 2076 : 2012 * | **6174 : 5735** * |

[1] CLEVER benchmark (with Ref 'C')

## 3.1 VERICODING DEFINITIONS

A *documentation* is a collection of natural language text that describes the *intent* (intended behavior) and optionally the *pseudocode* (prescribed algorithm) of some program. A *specification* (or *spec*) is a representation of the intent in a formal language. It contains both the function *signature* (name, input

types and output types) and the *conditions* (e.g., preconditions, postconditions) that the functions must satisfy. *Code* refers to a program implementation that can be executed or checked. A *proof* is a formal demonstration of the correctness of an implementation with respect to some specification. Specifications and code may refer to an external *context* that contains the definitions of the objects used. These can be boolean predicates, mathematical functions, data structures, and algorithms. The context may be defined within the same file or imported from external files. In ITPs, the specification signature and its implementation are grouped to form a *definition*. The implementation can be unfolded from the definition as needed, e.g. for the proof. The specification conditions and their proof are also paired to form a *theorem* where the proof details are opaque and cannot be unfolded from the theorem. By making proofs opaque, theorems can only depend on the conditions of other theorems and not on their proofs. In ATPs, the specification signature and its conditions are grouped together, while the implementation and proof interleave to form the (proof-carrying) code.

A *vericoding task* consists of the context and the spec, and optionally the documentation, which are passed to the LLM model. A *vericoding solution* contains the implementation, the proof, as well as any additional context that is needed, often in the form of imports, *helper* functions, and helper lemmas. As shown in Figure 1, we use tags to indicate different components of a task or solution file: documentation, spec, code, preamble, postamble and helper contexts. Most of the tasks –such as those from Verina, Clever, DafnyBench, and Numpy Triple– require vericoding for exactly one function. Other tasks, such as those from FVAPPS, require implementations of several functions and proofs that they work together in the desired way. For example, LF0007 (a lead-up to LF0023) asks for the `ins` and `pop` operations on a heap, and a proof that popping the elements off a heap produces a sorted list. Some Lean tasks include unit tests, which are put in the postamble. Examples of these tests can be found in FVAPPS, Verina, and Clever. Unit tests are not given to the LLM for vericoding, as the LLM can otherwise easily design implementations that pass all unit tests. They are instead applied to the implementation as an additional layer of checks after vericoding is completed.

**Lean (LB0000)**

```
-- <vc-preamble>
def valid_bitstr (v : List Int) : Prop :=
 ∀i, i < v.length → (v[i]? = some 0 ∨
    v[i]? = some 1)
def str2int (v : List Int) : Nat :=
 match v with
 | [] => 0
 | x :: xs => x.toNat + 2 * str2int xs
-- </vc-preamble>
-- <vc-helpers> ... -- </vc-helpers>
-- <vc-definitions>
def add (v1 v2 : List Int) : List Int :=
 sorry
-- </vc-definitions>
-- <vc-theorems>
theorem add_spec (v1 v2 : List Int)
 (h1 : valid_bitstr v1) (h2 : valid_bitstr
    v2) :
 valid_bitstr (add v1 v2) ∧
 str2int (add v1 v2) = str2int v1 + str2int
    v2 :=
 by sorry
-- </vc-theorems>
-- <vc-postamble> ... -- </vc-postamble>
```

**Verus (VB0000)**

```
// <vc-preamble>
use vstd::prelude::*;
verus! {
spec fn valid_bitstr(v: Seq<i8>) -> bool
{ forall |i: int| 0 <= i < v.len() ==> (v[i] == 0 || v[i] ==
    1)}
spec fn str2int(v: Seq<i8>) -> int
   decreases v.len()
{ if v.len() == 0 { 0 } else { v[0] +
   2 * str2int(v.subrange(1, v.len() as int)) } }
// </vc-preamble>
// <vc-helpers> ... // </vc-helpers>
// <vc-spec>
fn add(v1: &Vec<i8>, v2: &Vec<i8>) -> (result: Vec<i8>)
   requires valid_bitstr(v1@) && valid_bitstr(v2@)
   ensures valid_bitstr(result@),
      str2int(result@) == str2int(v1@) + str2int(v2@)
// </vc-spec>
// <vc-code>
{ assume(false); unreached() }
// </vc-code>
// <vc-postamble>
} fn main() {}
// </vc-postamble>
```

Figure 1: Examples of vericoding tasks. We use *vc* tags for elements introduced in Section 3.1.

## 3.2 ORIGINAL SOURCES AND TASK GENERATION

We generate vericoding tasks from three types of original sources (see bold text in Table 2) :

1. Formal verification or vericoding benchmarks such as DafnyBench (Loughridge et al., 2025) for Dafny, VerifiedCogen (JetBrains-Research, 2025) for Verus, and Verina (Ye et al., 2025) and Clever (Thakur et al., 2025) for Lean.

2. Vibe coding benchmarks such as APPS (Hendrycks et al., 2021), FVAPPS (Dougherty & Mehta, 2025) and HumanEval (Chen et al., 2021).

3. Documentations of mathematical function libraries such as Numpy (Harris et al., 2020, v2.3) and BigNum (Polubelova, 2022).

These sources often contain proposed implementations and sometimes proofs in addition to formal or informal specifications for each task. To obtain a vericoding task, we delete the implementation of the main coding task and all helper lemmas and proofs, and replace them with holes (e.g. `sorry`). For the vibe coding and documentation sources, we also perform autoformalization to obtain a formal spec for each task. We elaborate on our process for each source in Appendix 1.1.

## 3.3 SPEC TRANSLATIONS AND VALIDATION

$$
\begin{array}{l}
\hline
\mathsf{SpecTranslator}(\mathcal{L}_{\mathsf{source}}, \mathcal{L}_{\mathsf{target}}, \mathcal{S}, k, \mathcal{M}) \qquad\qquad \textit{// for specification } \mathcal{S}, \textit{ iterations } k, \textit{ and LLM } \mathcal{M} \\
\hline
\quad H \leftarrow \emptyset, \mathcal{P} \leftarrow \mathsf{BuildPrompt}(\mathcal{L}_{\mathsf{source}}, \mathcal{L}_{\mathsf{target}}, \mathcal{S}, \mathtt{translate}) \\
\quad \textbf{for } i \in \{1, \ldots, k\} \textbf{ do} : \quad \textit{// generate or fix translation, verify file in target language, and update history} \\
\qquad \mathcal{S}' \leftarrow \mathcal{M}(\mathcal{P}), (v, e) \leftarrow \mathsf{VerifyFile}(\mathcal{L}_{\mathsf{target}}, \mathcal{S}'), H \leftarrow H \cup \{\mathcal{S}', v, e\} \\
\qquad \textbf{if } v \textbf{ then return } (\mathcal{S}', \mathtt{success}) \textbf{ else } \mathcal{P} \leftarrow \mathsf{BuildPrompt}(\mathcal{L}_{\mathsf{source}}, \mathcal{L}_{\mathsf{target}}, \mathcal{S}, \mathtt{fix}, \mathcal{S}', H) \\
\quad \textbf{return } (\mathcal{S}', \mathtt{fail}) \\
\hline
\end{array}
$$

Figure 2: LLM-based translation of vericoding specifications.

To expand task coverage across languages, we translate tasks between Dafny, Verus, and Lean[1]. Figure 2 shows our LLM-based translation algorithm, which attempts up to $k$ iterations to produce verified translations. The process uses *structured processing* of tagged specifications (`vc-description`, `vc-preamble`, `vc-spec`, `vc-code`, `vc-postamble`) and a *conversational repair loop* that maintains translation history to learn from verification feedback. Since we translate only specifications (not implementations), we expect higher translation success rates as specifications are typically easier to translate than complete implementations. Translation counts are shown in Table 2 (plaintext).

**Validation:** We validated translated specifications through two approaches. First, we used the "LLM as a judge", asking it to compare each translation against its tagged source to verify faithful preservation of preconditions, postconditions, and invariants. This catches attempts to trivialize verification—such as replacing complex postconditions with `ensures true` or substituting function bodies with default values. Second, through manual inspection, we picked a random sample of translated specifications to inspect for correctness. We discovered that a handful of specifications admitted trivial solutions, which were due to lossy LLM transpilation or due to incomplete human-authored specifications that failed to fully capture their natural-language descriptions.

**Quality assessment:** We assess benchmark quality using a scoring framework that penalizes language-specific issues and near-duplicates. The quality score per file in a benchmark is $1 - (\sum_i w_i \times n_i)$ where $w_i$ is the normalized weight for issue type $i$ (weights sum to 1.0) and $n_i$ is the number of issue types $i$ in that file. Near-duplicate detection uses sentence transformers (Reimers & Gurevych, 2019) with FAISS indexing (Johnson et al., 2017) to compare normalized problem descriptions and specifications. Typical issues include Verus specs which return a default value instead of having an implementation (see Figure 7(a) in the Appendix for an example). All quality metrics, scores, and detected issues are embedded as metadata in the benchmark JSONL files. A summary of the quality metrics is shown in Table 5 in the Appendix.

## 4 VERICODING EXPERIMENTS

We quantify the vericoding success rate for LLMs such as GPT and Claude straight out of the box, without special techniques such as reinforcement learning or fine-tuning. Our experimental pipeline is simple: we present to the LLM a prompt that is annotated to indicate the context, the problem spec, as well as different holes (e.g. `sorry`'s) that require input from the LLM model. The model responds with blocks generated for each of the holes. The generated blocks are *validated* by checking for cheating patterns, and inserted into the task file template. The file is then *verified* with the proof checker. If both validation and verification pass, the LLM is deemed to have successfully

---

[1]With the exception of (FV)Apps/Humaneval where the source language is in Python. The translator from Python to Dafny extends on the one in Figure 2 with a generator-vs-judge game.

Table 3: Vericoding results

| | NumPyS | NumPy3 | DafnyBench | HumanEval | Verina | BigNum | VerifCogen | APPStest | Totals ↓ |
|---|---|---|---|---|---|---|---|---|---|
| **Dafny** | 58 tasks | 430 tasks | 443 tasks | 162 tasks | 157 tasks | 62 tasks | 172 tasks | 677 tasks | **2161 tasks** |
| **claude-opus-4.1** | 60.3% | 64.9% | 67.0% | 71.6% | 73.2% | 14.5% | 90.7% | 66.6% | **67.5%** |
| **gpt-5-mini** | 67.2% | 56.3% | 64.1% | 71.6% | 73.9% | 25.8% | 85.5% | 71.6% | **66.9%** |
| **gpt-5** | 65.5% | 56.3% | 63.9% | 72.2% | 76.4% | 19.4% | 86.6% | 69.1% | **66.1%** |
| **claude-sonnet-4** | 65.5% | 69.3% | 61.2% | 72.8% | 72.0% | 8.1% | 84.3% | 60.3% | **64.6%** |
| **gemini-2.5-pro** | 63.8% | 60.2% | 58.2% | 69.1% | 66.9% | 6.5% | 79.7% | 40.8% | **55.0%** |
| **grok-code** | 48.3% | 47.0% | 52.8% | 57.4% | 63.1% | 3.2% | 75.6% | 52.9% | **53.0%** |
| **glm-4.5** | 50.0% | 41.4% | 20.3% | 50.0% | 51.0% | 4.8% | 67.4% | 48.7% | **42.0%** |
| **gemini-2.5-flash** | 48.3% | 34.4% | 36.3% | 42.6% | 45.9% | 0.0% | 57.0% | 36.9% | **38.2%** |
| **deepseek-chat-v3.1** | 39.7% | 43.0% | 30.2% | 45.1% | 46.5% | 3.2% | 50.0% | 30.6% | **36.2%** |
| **model union** | **75.9%** | **77.9%** | **71.9%** | **93.2%** | **87.3%** | **48.4%** | **95.9%** | **83.0%** | **82.2%** |
| **Verus** | 58 tasks | 581 tasks | 443 tasks | 161 tasks | 156 tasks | 62 tasks | 172 tasks | 536 tasks | **2166 tasks** |
| **gpt-5** | 18.9% | 47.5% | 18.3% | 15.5% | 30.7% | 3.2% | 49.4% | 26.5% | **30.9%** |
| **claude-opus-4.1** | 18.9% | 29.7% | 19.2% | 9.9% | 26.2% | 1.6% | 63.4% | 18.1% | **24.6%** |
| **gemini-2.5-pro** | 34.5% | 31.5% | 19.7% | 9.3% | 25.6% | 1.6% | 52.9% | 16.0% | **24.1%** |
| **claude-sonnet-4** | 20.7% | 22.0% | 19.0% | 5.6% | 27.5% | 1.6% | 48.2% | 13.4% | **19.9%** |
| **glm-4.5** | 8.6% | 23.0% | 11.3% | 5.6% | 16.0% | 0.0% | 27.3% | 16.8% | **16.6%** |
| **gpt-5-mini** | 3.4% | 21.3% | 9.5% | 6.2% | 24.3% | 0.0% | 22.6% | 19.0% | **16.5%** |
| **grok-code** | 10.3% | 22.9% | 10.6% | 3.1% | 17.9% | 1.6% | 25.0% | 13.8% | **15.5%** |
| **gemini-2.5-flash** | 10.3% | 16.3% | 7.7% | 1.2% | 12.8% | 0.0% | 16.2% | 5.2% | **9.8%** |
| **deepseek-chat-v3.1** | 3.4% | 6.9% | 5.0% | 1.8% | 10.2% | 0.0% | 8.7% | 6.5% | **6.1%** |
| **model union** | **37.9%** | **55.8%** | **34.8%** | **26.1%** | **46.8%** | **4.8%** | **77.9%** | **38.8%** | **44.3%** |
| **Lean** | 59 tasks | 603 tasks | 440 tasks | 161 tasks[1] | 189 tasks | 62 tasks | 172 tasks | 675 tasks | **2361 tasks** |
| **gpt-5** | 45.8% | 5.1% | 32.0% | 3.7% | 14.3% | 12.9% | 34.9% | 18.4% | **17.9%** |
| **claude-sonnet-4** | 30.5% | 0.7% | 11.4% | 1.2% | 13.8% | 1.6% | 10.5% | 24.0% | **11.9%** |
| **gemini-2.5-pro** | 23.7% | 0.3% | 14.1% | 3.1% | 12.2% | 1.6% | 9.3% | 19.9% | **10.9%** |
| **claude-opus-4.1** | 20.3% | 1.0% | 11.8% | 3.1% | 15.3% | 1.6% | 10.5% | 19.3% | **10.7%** |
| **gpt-5-mini** | 15.3% | 0.3% | 15.0% | 0.0% | 1.6% | 1.6% | 4.1% | 6.8% | **5.7%** |
| **grok-code** | 10.2% | 0.0% | 9.1% | 0.0% | 1.1% | 0.0% | 4.7% | 6.2% | **4.2%** |
| **glm-4.5** | 6.8% | 0.2% | 2.5% | 0.0% | 0.5% | 0.0% | 1.7% | 2.7% | **1.6%** |
| **gemini-2.5-flash** | 0.0% | 0.2% | 0.5% | 0.0% | 0.0% | 0.0% | 0.6% | 1.9% | **0.7%** |
| **deepseek-chat-v3.1** | 0.0% | 0.0% | 0.0% | 0.0% | 0.0% | 0.0% | 0.0% | 0.1% | **0.0%** |
| **model union** | **52.5%** | **7.6%** | **34.1%** | **8.1%** | **25.4%** | **12.9%** | **44.2%** | **38.5%** | **26.8%** |

[1] From the CLEVER benchmark

**Additional results**

| | gpt-5 | gemini-pro | gpt-mini | opus | grok-c | sonnet | glm | deepseek | gemini-fl | union |
|---|---|---|---|---|---|---|---|---|---|---|
| **(A)** 4006 tasks | 38.0% | 12.7% | 6.2% | 6.1% | 4.5% | 4.0% | 1.7% | 0.1% | 0.1% | **41.8%** |
| **(B)** 782 tasks | 73.8% | 78.5% | | 89.2% | 82.1% | 86.1% | 86.8% | 74.3% | | **96.0%** |
| | (715) | (339) | | (782) | (782) | (703) | (182) | (404) | | |
| **(C)** 157 tasks | | | | | | | | | | |
| **Dafny** | 76.4% | 66.8% | 73.9% | 73.2% | 63.0% | 72.0% | 50.9% | 46.4% | 45.9% | **87.3%** |
| **Dafny +vibe** | 68.8% | 68.8% | 72.0% | 66.9% | 58.6% | 70.1% | 51.6% | 42.7% | 49.0% | **86.6%** |
| **Verus** | 30.7% | 25.6% | 24.3% | 26.2% | 17.9% | 27.5% | 16.0% | 10.2% | 12.8% | **46.8%** |
| **Verus +vibe** | 22.4% | 25.6% | 17.9% | 22.4% | 18.5% | 23.7% | 12.8% | 10.2% | 18.5% | **37.8%** |

**(A)** FVAPPS in Lean    **(B)** DafnyBench verification    **(C)** Spec+vibe vericoding for Verina

```
Vericoder(L, S, P, k, M)        // for L ∈ {dafny, verus, lean}, spec S, prompt templates P, iterations k, and LLM M
─────────────────────────────────────────────────────────────────────────────────────────
Q ← FillPrompt(L, S, P, generate)        // the prompt for the first iteration is to generate code and proofs
for i ∈ {1, ..., k} do :                  // iterate until validation and verification succeeds, or until max iterations reached
    B₁, ..., Bₙ ← M(Q), F ← S[B₁, ..., Bₙ]    // prompt M and reconstruct the file F from S and generated blocks
    (v₀, w₀) ← ValidateBlocks(L, B₁, ..., Bₙ)    // check for verification bypass patterns in generated blocks
    (v₁, w₁) ← VerifyFile(L, F)            // run dafny, verus or lean to verify the generated file
    if v₀ ∧ v₁ then return (success, F)    // else we have validation or verification errors
    Q ← FillPrompt(L, S, P, fix, F, w₀, w₁)    // we pass any error messages w₀, w₁ to M at the next iteration
return (fail, F)                          // in case of failure after max iterations, we return the last generated file
```

Figure 3: Vericoding process using an LLM model $M$ for vericoding tasks $S$ and language $L$

completed the task. Otherwise, the error messages are passed to the LLM for correction. We allow only a fixed number of such iterations before deciding that the LLM has failed. This process is described more precisely in Figure 3. The validation tests are specific for each proof language. They depend on corresponding compile options, syntax and programming patterns. We designed the tests by taking into account known proof bypass patterns (e.g. using *sorry* for Lean), as well as various LLM cheating strategies that we have identified during our experiments.

**Prompts and hyperparameters:** We have two prompt templates for each language, listed in Appendix 1.3: one for code and proof generation and one for fixing verification errors in previously generated files. We informally explored prompting variations, but did not optimize the prompts fully. The Verus prompt might benefit from examples of common lemmas in vstd. For Dafny/Verus, we gave sample syntax in the prompt. Each LLM had 5 attempts per task, except 10 attempts on the challenging BigNum dataset. In comparison, Harmonic used half a million CPUs to run Lean for its IMO work (Harmonic, 2025).

**LLM cheating detection:** Our block validation script systematically detects potential LLM cheating patterns that could bypass the proof checker, possibly by changing the goals. These include : (1) Using `assume(false)` or `sorry` to disable the proof checker. We can instruct the proof checker to reject such proofs. (2) Changing postconditions in the spec to `ensures true` to make it trivial to prove. We can block the LLM from altering the specs. (3) Implementation leakage from the spec. We use ghost functions in Dafny/Verus to partly mitigate this. There will typically be some level of implementation leakage from the specs–if not the full implementation, then at least the spirit of it. Indeed, we expect LLMs to do so, so the onus is on spec writers to create good specs (see, e.g., the CLEVER benchmark). (4) We anticipated a cunning form of cheating where the LLM begins a comment section in one block and closes the comment section in another, effectively erasing any intermediate task spec through this trick. Thankfully, this did not happen.

**Manual inspection:** To quantify the validity of the vericoding process, we perform manual inspection of 5 randomly chosen successful vericoding outputs for each language and data source, noting any LLM behavior that could be considered cheating. The reports, detailed in the supplementary material, show language-specific patterns: Dafny exhibited no issues (except for redundant lemmas), with LLMs correctly giving trivial solutions when the specs were weak. Verus showed many weak specs due to spec translation issues, e.g. deviations from the original BigNum specs in Dafny. Lean displayed occasional redundant lemma additions during vericoding, and had some weak specs from the original sources. Across Dafny, Verus and Lean, conditioned on vericoding success, roughly 9% of the specs were too weak and another 15% had poor translations. Note that these problematic specs still make perfectly valid vericoding tasks, just different tasks than originally intended. No further cheating was discovered, other than those which were caught by our validation checks.

## 4.1 VERICODING RESULTS

Table 3 shows that LLM vericoding worked best on Dafny (82.2%), followed by Verus (44.2%) and Lean (26.8%). Claude-Opus- 4.1 excelled at Dafny, while GPT-5 led on Verus and Lean. The "model union" numbers denote the fraction solved by at least one of the LLMs.

Verus success rates are probably lower due to several factors: unlike Dafny's uniform mathematical types, Verus distinguishes between ghost types (for specifications) and Rust native types (for execution), requiring verification of machine-level complexities such as overflow handling. Additionally, LLM translations often fail to systematically map between these type systems, and Verus is a newer, lower-resource language than Dafny.

Lean's lower success probably stems mostly from LLMs being trained primarily on mathematical theorem proving rather than code verification. The Lean prompt might benefit from examples of how to use new tactics such as `grind` and `canonical`. ITPs and ATPs will also require different strategies for exploiting LLMs: A human writing Lean gets constant feedback about the proof state, while our scaffolding only told the LLM what the proof state was 5 times per example.

We have separated FVAPPS into Row A of Table 3 because the specs are weaker and allow for more trivial solutions. We did not apply the unit tests associated to the tasks in these experiments. Here, GPT-5 is again seen to perform best.

**Verification alone:** We use the original dataset from Loughridge et al. (2025) with 782 tasks and its original prompts to gauge how much the LLMs have improved on verification. Table 3 shows the success rates (sample sizes in parentheses) for different LLMs, revealing rapid progress: The June 2024 state-of-the-art of 68% with Opus-3 has now risen to 89% with Opus-4.1 and 96% for the model union. See Appendix 1.6 for more details.

**Hoare triples in Lean:** The model union achieved 7.6% success on our novel Numpy-Triple benchmark, which is below the language average of 26.8%. This is likely due to the `mvcgen` feature being new. We expect the success rate to improve significantly over the next two years when newer LLMs are trained on datasets that use this feature and as SMT-solver-based tactics in Lean become more proficient at tackling the proof obligations.

**Spec and vibe coding:** In addition to formal specifications, several of our source benchmarks contain informal descriptions of vericoding tasks. One of these is Verina, for which we perform a separate set of experiments including the informal descriptions in the LLM prompt. The aim is to test their effect on vericoding performance. We find that including the "vibe" information provides no statistically significant performance improvement (indeed, the results seen to be slightly worse on average), so we decided not to extended this experiment to our full benchmark, and plan to study this more extensively in future work.

**Improvements from ensemble methods:** Results from taking the union of successful attempts from different models can be found in Table 3. This potentially suggests that in the future, vericoding tasks should be decomposed into smaller parallel tasks that can be tackled by different models. To make this like a mixture of experts (MoE) strategy in advanced language models (Jiang et al., 2024), we would need to implement a router model (perhaps just a pre-deep-learning bag-of-words model) that decides which LLM will attempt a problem.

## 4.2 PARAMETERS INFLUENCING VERICODING DIFFICULTY

We investigate three different parameters that one may expect to impact vericoding complexity: *spec length* (the number of characters in the spec source code), *solution length* (the number of characters in the LLM's solution), and *spec ratio* (code length divided by spec length). Note that an LLM's solution may or may not be code that satisfies the spec or even parses in its target language, since not all solutions are correct.

Spec length depends on the number of preconditions, postconditions, and helper definitions. More preconditions ease verification by providing assumptions, while more postconditions increase difficulty by requiring additional proofs. Helper definitions create interdependent proof obligations that are harder to solve.

We find that LLMs easily generate implementations (consistent with vibe coding success), but struggle more with proofs — adding invariants/assertions for ATPs or choosing tactics for ITPs. Our results, shown in Figure 4, exclude all cases where the LLM fails to propose an implementation.

Figure 4 shows the relation between vericoding success and the various parameters. Spec length is the weakest predictor of vericoding complexity, presumably because simple problems can have lengthy solutions (e.g., Fermat's Last Theorem). Additionally, longer specs often contain helper

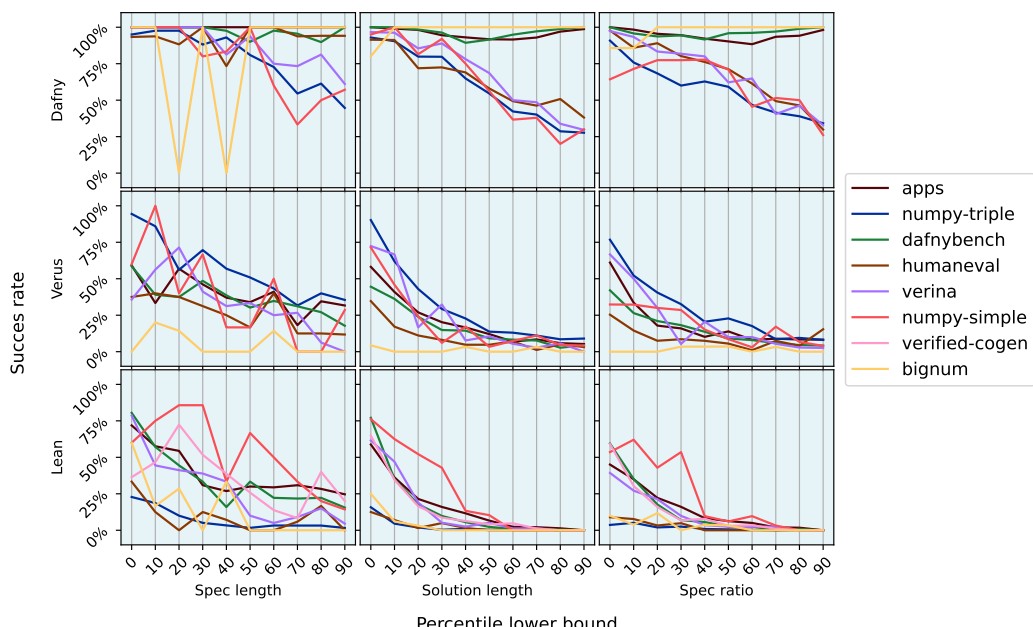

Figure 4: Vericoding success as a function of task spec length (top row), generated code length (middle row) and spec ratio (bottom row) which is the code length divided by the spec length. **sorted by size**

functions that do not require vericoding, unlike additional postconditions which increase difficulty. Solution length shows a clearer trend: longer implementations are more likely incorrect, similar to human coding where more code increases error probability. With finite iterations per task, shorter implementations allow better use of each iteration. Spec ratio trends fall between these patterns, following solution length trends more closely.

## 5 CONCLUSIONS

We have made the case for *vericoding* as a rigorous alternative to vibe coding, released by far the largest vericoding benchmark to date, and demonstrated success rates ranging from 27% for Lean to 82% for Dafny. The rapid rate of LLM progress (improving formal verification success from 68% to 96% in just over a year) appears likely to futher improve vericoding success in the near future. As AI-generated code becomes more widely deployed, formal verification will become increasingly critical for ensuring code correctness.

Our contributions complement concurrent efforts such as cslib (Montesi et al., 2025), which provides additional Lean tasks that we can now approach with LLM-generated solutions.

We see many opportunities for improving our results. While the tasks in our dataset are typically solvable in under 100 lines of code, extending formal specs to more complex benchmarks like SWE-bench (Jimenez et al., 2024) or BashBench (Bhatt et al., 2025) presents exciting future challenges. Our current results, achieved without extensive prompt optimization, also suggest significant room for improvement through more advanced techniques such as tree search and reinforcement learning approaches, e.g., see Seed-Prover (Chen et al., 2025). Future work could also explore networks of collaborative LLMs, each leveraging their unique strengths to tackle verification problems.

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

# A APPENDIX

Our Dafny, Lean, Verus benchmarks can be found in our Supplementary Material. In particular, the file `benchmarks/vericoding_benchmark_v1.csv` contains a summary of the metadata for all 12,504 tasks. The file `experiments/vericoding_results_v1.csv` contains the outcomes of all 55,397 vericoding experiments, one for each task and each language model.

## 1.1 ORIGINAL SOURCES

*DafnyBench.* In this source, besides removing existing code and proofs to create a vericoding task, we split the larger files (which contain multiple methods, code and lemmas) into smaller tasks, each with one method and its dependencies. We also filter out files that do not contain any methods.

*Verified Cogen.* This benchmark from JetBrains Research contains Rust programs with Verus specifications focusing on memory safety and functional correctness. To create tasks, we removed implementation bodies.

*Verina and Clever.* The former gathers problems on data structures, algorithms, and mathematical properties, while the latter focuses on functional correctness of algorithms and data structures. We removed all spec generation and spec isomorphism requirements from the tasks.

*APPS and HumanEval.* These are popular coding benchmarks containing Python problems with natural language specifications and test cases. For APPS, we focused on the 5,000 instances in the test split (Hendrycks et al., 2021), as these have more tests per solution and generally harder tasks. We generated Dafny specifications by means of a specific LLM translator (see Section 3.3).

*FVAPPS.* It translates the *Train* split of the APPS Python benchmark to Lean and adds formal specs. We keep the unit tests for each task and apply them in our vericoding experiments.

*NumPy simple and NumPy triple.* (Harris et al., 2020). Scraped from numpy docs, checked by hand (we need to recheck considering some specs have slipped past), simple format and triple format. Represent specs as Hoare triples, with preconditions, the program and the postcondition, using new *mvcgen* feature in Lean.

*BigNum.* This is a collection of arithmetic algorithms on big numbers in Dafny that we wrote from scratch: simplified versions of 13 functions often used in cryptography which is a prime target for formal verification (Barbosa et al., 2021). By varying the combination of helper functions included, we obtain 62 tasks.

## 1.2 LICENSES

Our code/datasets are released under the MIT License. Table 4 shows dataset sources and licenses.

| Dataset | Source License | Source URL |
|---|---|---|
| APPS (test) | MIT | github.com/hendrycks/apps |
| DafnyBench | Apache 2.0 | github.com/sun-wendy/DafnyBench |
| NumPy Triple | BSD-3-Clause | github.com/numpy/numpy |
| verified-cogen | Permission requested | github.com/JetBrains-Research/verified-cogen |
| Verina | Apache 2.0 | github.com/sunblaze-ucb/verina |
| Bignum | MIT | Our own work |
| NumPy Simple | BSD-3-Clause | github.com/numpy/numpy |
| HumanEval Python | MIT | github.com/openai/human-eval |
| FVAPPS | MIT | huggingface.co/datasets/quinn-dougherty/fvapps |
| Clever | MIT | github.com/trishullab/clever |

Table 4: Licenses of dataset sources used in our benchmark

## 1.3 PROMPTS AND HYPERPARAMETERS

The following prompts are used for code generation and verification repair in different languages.

Each LLM had five attempts at a sample: One initial attempt, and four retries with the error message from applying the proof checker to the previous attempt. We did not run multiple trials with a fresh history, but it may be worth looking into whether an LLM gets locked into a bad approach. For the bignums dataset, we instead gave the LLM one initial attempt and nine retries.

**Dafny code generation prompt:**

```
CRITICAL: Respond with ONLY a JSON array. No explanations, reasoning, or markdown. Start with [ and end with
    ].

The task is to generate implementations for `<vc-code>` and `<vc-helpers>` sections in a Dafny file.

TURN 1 of {max_iterations}: This is the initial code generation phase. You have {max_iterations} total turns
    to get this right, so you can iterate and improve.

INPUT: a Dafny file containing {placeholder_count} placeholder sections (`<vc-code>` and/or `<vc-helpers>`
    tags) that need to be filled in.

OUTPUT: Return a JSON array with EXACTLY {placeholder_count} replacements (one for each placeholder section in
     the file), in order from top to bottom:
```json
["function min(a: int, b: int): int {{ if a < b then a else b }}", "{{\n result := ComputeResult(n, pos);\n
    }}"]
```

SECTION-SPECIFIC RULES:

**For `<vc-helpers>` sections:**
- Provide COMPLETE helper function/lemma definitions ONLY
- Each helper should be a standalone function/predicate/lemma with proper signature
- Example: `function min(a: int, b: int): int {{ if a < b then a else b }}`
- Example: `predicate IsValid(x: int) {{ x >= 0 }}`
- Example: `lemma HelperLemma(x: int) ensures x + 0 == x {{ }}`
- DO NOT include method body code, variable assignments, or code fragments
- DO NOT include opening/closing braces unless they're part of the function body

**For `<vc-code>` sections:**
- Provide method body implementation code ONLY
- Always include opening/closing braces: `{{\n \n}}`
- Example: `{{\n result := min(a, b) + 1;\n}}`
- This is where you call helper functions and implement the main logic
- Include variable declarations, assignments, and control flow

CRITICAL RULES:
- The ORIGINAL file contains EXACTLY {placeholder_count} placeholder sections - your JSON array must have
    EXACTLY {placeholder_count} elements
- Provide exactly one replacement for each placeholder section in the file, in the exact order they appear (
    top to bottom)
- Each replacement should be the exact code that will replace everything between the tags
- NEVER use verification bypasses: `{{:axiom}}`, `assume` statements, or other verification shortcuts
- Implement actual proofs and logic instead of bypassing verification
- Use valid Dafny syntax for all implementations
- Satisfy all `requires` and `ensures` clauses from the method/function specifications
- Do not add trivial or unnecessary annotations
- Return ONLY a valid JSON array, no explanations or markdown
- DO NOT include any reasoning, explanations, or commentary
- DO NOT use markdown code blocks around the JSON
- Your response must start with [ and end with ]
- Each JSON string must be properly escaped with double quotes

CRITICAL: Your entire response must be ONLY the JSON array. Any text before or after the JSON will cause
    parsing failure.

CRITICAL: Do NOT use `assume {{:axiom}}`, `assume`, or any verification bypasses. Implement real logic!

DAFNY FILE WITH PLACEHOLDER SECTIONS:
{code}
```

**Dafny verification repair prompt:**

```
CRITICAL: Respond with ONLY a JSON array. No explanations, reasoning, or markdown. Start with [ and end with
    ].

The task is to fix implementations in `<vc-code>` and `<vc-helpers>` sections that failed verification.

TURN {iteration} of {max_iterations}: You are making progress and have multiple turns to iterate and improve
    your implementation.

INPUT: The ORIGINAL file contains {placeholder_count} placeholder sections (`<vc-code>` and/or `<vc-helpers>`
    tags) that need to be fixed based on verification errors.

OUTPUT: Return a JSON array with EXACTLY {placeholder_count} fixed replacements (one for each placeholder
    section in the ORIGINAL file), in order from top to bottom:
```json
["function min(a: int, b: int): int {{ if a < b then a else b }}", "{{\n result := min(a, b) + 1;\n}}"]
```

SECTION-SPECIFIC RULES:
```

```
**For `<vc-helpers>` sections:**
- Provide COMPLETE helper function/lemma definitions ONLY
- Each helper should be a standalone function/predicate/lemma with proper signature
- Example: `function min(a: int, b: int): int {{ if a < b then a else b }}`
- Example: `predicate IsValid(x: int) {{ x >= 0 }}`
- Example: `lemma HelperLemma(x: int) ensures x + 0 == x {{ }}`
- DO NOT include method body code, variable assignments, or code fragments
- DO NOT include opening/closing braces unless they're part of the function body
- Add comment `/* helper modified by LLM (iteration {iteration}): [brief description] */` before modified
    helpers

**For `<vc-code>` sections:**
- Provide method body implementation code ONLY
- Always include opening/closing braces: `{{\n \n}}`
- Example: `{{\n result := min(a, b) + 1;\n}}`
- This is where you call helper functions and implement the main logic
- Include variable declarations, assignments, and control flow
- Add comment `/* code modified by LLM (iteration {iteration}): [brief description] */` at the start of method
      body

CRITICAL RULES:
- The ORIGINAL file contains EXACTLY {placeholder_count} placeholder sections – your JSON array must have
      EXACTLY {placeholder_count} elements
- Provide exactly one replacement for each placeholder section in the file, in the exact order they appear (
      top to bottom)
- Each replacement should be the exact fixed code that will replace everything between the tags
- PRIORITY: If the error is a compilation error (syntax, type, resolution errors), fix it first before
      addressing verification issues
- NEVER use verification bypasses: `{{:axiom}}`, `assume` statements, or other verification shortcuts
- Implement actual proofs and logic instead of bypassing verification
- Use valid Dafny syntax for all implementations
- Satisfy all `requires` and `ensures` clauses from the method/function specifications
- Do not add trivial or unnecessary annotations
- Return ONLY a valid JSON array, no explanations or markdown
- DO NOT include any reasoning, explanations, or commentary
- DO NOT use markdown code blocks around the JSON
- Your response must start with [ and end with ]
- Each JSON string must be properly escaped with double quotes

CRITICAL: Your entire response must be ONLY the JSON array. Any text before or after the JSON will cause
      parsing failure.

CRITICAL: Do NOT use `assume {{:axiom}}`, `assume`, or any verification bypasses. Implement real logic!

ERROR DETAILS from Dafny verification:
{errorDetails}

ORIGINAL FILE (for context):
{original_code}

CURRENT ITERATION FILE (with failed implementations to learn from):
{code}
```

## Verus code generation prompt:

```
CRITICAL: Respond with ONLY a JSON array. No explanations, reasoning, or markdown. Start with [ and end with
      ].

The task is to generate implementations for `<vc-code>` and `<vc-helpers>` sections in a Verus file.

TURN 1 of {max_iterations}: This is the initial code generation phase. You have {max_iterations} total turns
      to get this right, so you can iterate and improve.

INPUT: a Verus file containing {placeholder_count} placeholder sections (`<vc-code>` and/or `<vc-helpers>`
      tags) that need to be filled in.

OUTPUT: Return a JSON array with EXACTLY {placeholder_count} replacements (one for each placeholder section in
      the file), in order from top to bottom:
```json
["fn min(a: int, b: int) -> int {{ if a < b {{ a }} else {{ b }} }}", "{{\n let result = min(a, b);\n result\n
      }}"]
```

SECTION-SPECIFIC RULES:

**For `<vc-helpers>` sections:**
- Provide COMPLETE helper function/lemma definitions ONLY
- Each helper should be a standalone function/predicate/lemma with proper signature
- Example: `fn min(a: int, b: int) -> int {{ if a < b {{ a }} else {{ b }} }}`
- Example: `spec fn is_valid(x: int) -> bool {{ x >= 0 }}`
- Example: `proof fn helper_lemma(x: int) ensures x + 0 == x {{ }}`
- DO NOT include method body code, variable assignments, or code fragments
- DO NOT include opening/closing braces unless they're part of the function body

**For `<vc-code>` sections:**
- Provide method/function body implementation code ONLY
- Always include opening/closing braces: `{{\n \n}}`
- Example: `{{\n let result = min(a, b);\n result\n}}`
- This is where you call helper functions and implement the main logic
- Include variable declarations, assignments, and control flow
```

```
CRITICAL RULES:
- The ORIGINAL file contains EXACTLY {placeholder_count} placeholder sections - your JSON array must have
    EXACTLY {placeholder_count} elements
- Provide exactly one replacement for each placeholder section in the file, in the exact order they appear (
    top to bottom)
- Each replacement should be the exact code that will replace everything between the tags
- NEVER use verification bypasses: `assume` statements, `unimplemented!()`, or other verification shortcuts
- Implement actual proofs and logic instead of bypassing verification
- Use valid Verus/Rust syntax for all implementations
- Use proper Verus syntax: `requires`, `ensures`, `invariant`, `decreases` (without parentheses)
- IMPORTANT: Use each of `requires`, `ensures`, and `invariant` AT MOST ONCE per item
  - List multiple conditions as comma-separated entries on separate lines
  - Do NOT repeat `requires`/`ensures`/`invariant` keywords for additional lines
  - Prefer commas at line ends in spec blocks; avoid semicolons
- While-loop header style (single invariant block):

  while CONDITION
     invariant
         INV1,
         INV2,
     decreases MEASURE
  {{
     // body
  }}

- Function spec header style (single requires/ensures blocks):

  fn f(..) -> T
     requires
         PRE1,
         PRE2,
     ensures
         POST1,
         POST2,
  {{
     // body
  }}
- Use Verus types like `nat`, `int`, `Vec<T>`, `Seq<T>`, etc.
- Use `@` for sequence/vector indexing when needed (e.g., `v@[i]`)
- Use proof blocks with `proof {{ ... }}` when necessary
- Return ONLY a valid JSON array, no explanations or markdown
- DO NOT include any reasoning, explanations, or commentary
- DO NOT use markdown code blocks around the JSON
- Your response must start with [ and end with ]
- Each JSON string must be properly escaped with double quotes

CRITICAL: Your entire response must be ONLY the JSON array. Any text before or after the JSON will cause
    parsing failure.

CRITICAL: Do NOT use `assume`, `unimplemented!()`, or any verification bypasses. Implement real logic!

VERUS FILE WITH PLACEHOLDER SECTIONS:
{code}
```

## Verus verification repair prompt:

```
CRITICAL: Respond with ONLY a JSON array. No explanations, reasoning, or markdown. Start with [ and end with
    ].

The task is to fix implementations in `<vc-code>` and `<vc-helpers>` sections that failed verification.

TURN {iteration} of {max_iterations}: You are making progress and have multiple turns to iterate and improve
    your implementation.

INPUT: The ORIGINAL file contains {placeholder_count} placeholder sections (`<vc-code>` and/or `<vc-helpers>`
    tags) that need to be fixed based on verification errors.

OUTPUT: Return a JSON array with EXACTLY {placeholder_count} fixed replacements (one for each placeholder
    section in the ORIGINAL file), in order from top to bottom:
```json
["fn min(a: int, b: int) -> int {{ if a < b {{ a }} else {{ b }} }}", "{{\n let result = min(a, b);\n result\n
    }}"]
```

SECTION-SPECIFIC RULES:

**For `<vc-helpers>` sections:**
- Provide COMPLETE helper function/lemma definitions ONLY
- Each helper should be a standalone function/predicate/lemma with proper signature
- Example: `fn min(a: int, b: int) -> int {{ if a < b {{ a }} else {{ b }} }}`
- Example: `spec fn is_valid(x: int) -> bool {{ x >= 0 }}`
- Example: `proof fn helper_lemma(x: int) ensures x + 0 == x {{ }}`
- DO NOT include method body code, variable assignments, or code fragments
- DO NOT include opening/closing braces unless they're part of the function body
- Add comment `/* helper modified by LLM (iteration {iteration}): [brief description] */` before modified
    helpers

**For `<vc-code>` sections:**
- Provide method/function body implementation code ONLY
- Always include opening/closing braces: `{{\n \n}}`
- Example: `{{\n let result = min(a, b);\n result\n}}`
```

```
- This is where you call helper functions and implement the main logic
- Include variable declarations, assignments, and control flow
- Add comment '/* code modified by LLM (iteration {iteration}): [brief description] */' at the start of method
      body

CRITICAL RULES:
- The ORIGINAL file contains EXACTLY {placeholder_count} placeholder sections - your JSON array must have
      EXACTLY {placeholder_count} elements
- Provide exactly one replacement for each placeholder section in the file, in the exact order they appear (
      top to bottom)
- Each replacement should be the exact fixed code that will replace everything between the tags
- PRIORITY: If the error is a compilation error (syntax, type, resolution errors), fix it first before
      addressing verification issues
- Use proper Verus syntax: 'requires', 'ensures', 'invariant', 'decreases' (without parentheses)
- IMPORTANT: Use each of 'requires', 'ensures', and 'invariant' AT MOST ONCE per item
  - List multiple conditions as comma-separated entries on separate lines
  - Do NOT repeat 'requires'/'ensures'/'invariant' keywords for additional lines
  - Prefer commas at line ends in spec blocks; avoid semicolons
- While-loop header style (single invariant block):

  while CONDITION
      invariant
          INV1,
          INV2,
      decreases MEASURE
  {{
      // body
  }}

- Function spec header style (single requires/ensures blocks):

  fn f(..) -> T
      requires
          PRE1,
          PRE2,
      ensures
          POST1,
          POST2,
  {{
      // body
  }}
- Use proof blocks with 'proof {{ ... }}' for complex proofs
- Use 'assert()' statements within proof blocks for intermediate steps
- Use Verus types and operators ('nat', 'int', 'Vec<T>', 'Seq<T>', '@', etc.)
- NEVER use verification bypasses: 'assume' statements, 'unimplemented!()', or other verification shortcuts
- Implement actual proofs and logic instead of bypassing verification
- Use '@' for sequence/vector indexing when needed (e.g., 'v@[i]')
- Return ONLY a valid JSON array, no explanations or markdown
- DO NOT include any reasoning, explanations, or commentary
- DO NOT use markdown code blocks around the JSON
- Your response must start with [ and end with ]
- Each JSON string must be properly escaped with double quotes

CRITICAL: Your entire response must be ONLY the JSON array. Any text before or after the JSON will cause
      parsing failure.

CRITICAL: Do NOT use 'assume', 'unimplemented!()', or any verification bypasses. Implement real logic!

ERROR DETAILS from Verus verification:
{errorDetails}

ORIGINAL FILE (for context):
{original_code}

CURRENT ITERATION FILE (with failed implementations to learn from):
{code}
```

## Lean code generation prompt:

```
The task is to generate implementations and proofs for "sorry" placeholders and '<vc-helpers>' sections in a
      Lean file.

TURN 1 of {max_iterations}: This is the initial code generation phase. You have {max_iterations} total turns
      to get this right, so you can be strategic about using "sorry" temporarily in early iterations.

INPUT: a Lean file containing {placeholder_count} placeholder sections ("sorry" keywords and/or '<vc-helpers>'
      tags) in place of desired implementations, proofs, and helper code.

OUTPUT: Return a JSON array with EXACTLY {placeholder_count} replacements (one for each placeholder in the
      file), in order from top to bottom:
```json
["first_implementation", "helper_code", "second_proof", "third_implementation"]
```

CRITICAL RULES:
- The ORIGINAL file contains EXACTLY {placeholder_count} placeholder sections - your JSON array must have
      EXACTLY {placeholder_count} elements
- Provide exactly one replacement for each placeholder in the file, in the exact order they appear (top to
      bottom)
- For "sorry" placeholders: Each replacement should be the exact code/proof that goes where "sorry" appears
- For '<vc-helpers>' sections: Provide helper definitions, theorems, lemmas, or utility code
```

```
- IMPORTANT: Your replacement text goes directly where the placeholder is - if you need helper functions for "
    sorry", define them inline within the replacement
- RESTRICTED AREAS: You CANNOT modify or replace "sorry" keywords that appear inside `<vc-preamble>` sections
    - these are protected and not counted in the {placeholder_count}
- AVOID verification bypasses: "sorry", "admit", "axiom", "unsafe", "Unchecked.cast", or "@[extern]"
- You may use "sorry" temporarily in early iterations to get error feedback, but must remove all verification
    bypasses for final success
- Use valid Lean syntax for all replacements
- You may include helper definitions, theorems, and lemmas in replacements when needed
- For helper definitions added within a replacement, add comment -- LLM HELPER before them
- Return ONLY a valid JSON array, no explanations or markdown

LEAN FILE WITH PLACEHOLDERS:
{code}
```

## Lean verification repair prompt:

```
The task is to fix implementations and proofs that failed verification by providing replacements for any
    remaining "sorry" placeholders and `<vc-helpers>` sections.

TURN {iteration} of {max_iterations}: You are making progress - this gives you room to iterate and improve.
    Use "sorry" strategically if needed in intermediate turns.

INPUT: The ORIGINAL file contains {placeholder_count} placeholder sections ("sorry" keywords and/or `<vc-
    helpers>` tags) that need to be replaced based on verification errors.

OUTPUT: Return a JSON array with EXACTLY {placeholder_count} fixed replacements (one for each placeholder in
    the ORIGINAL file), in order from top to bottom:
```json
["fixed_implementation", "fixed_helper_code", "fixed_proof", "another_fixed_implementation"]
```

CRITICAL RULES:
- The ORIGINAL file contains EXACTLY {placeholder_count} placeholder sections - your JSON array must have
    EXACTLY {placeholder_count} elements
- Provide exactly one replacement for each placeholder in the file, in the exact order they appear (top to
    bottom)
- For "sorry" placeholders: Each replacement should be the exact fixed code/proof that goes where "sorry"
    appears
- For `<vc-helpers>` sections: Provide fixed helper definitions, theorems, lemmas, or utility code
- IMPORTANT: Your replacement text goes directly where the placeholder is - if you need helper functions for "
    sorry", define them inline within the replacement
- RESTRICTED AREAS: You CANNOT modify or replace "sorry" keywords that appear inside `<vc-preamble>` sections
    - these are protected and not counted in the {placeholder_count}
- MINIMIZE verification bypasses: "sorry", "admit", "axiom", "unsafe", "Unchecked.cast", or "@[extern]"
- You may use "sorry" temporarily to get error feedback, but each iteration should remove more verification
    bypasses
- Use valid Lean syntax for all replacements
- You may include helper definitions, theorems, and lemmas in replacements when needed
- For helper definitions added within a replacement, add comment -- LLM HELPER before them
- Return ONLY a valid JSON array, no explanations or markdown

ITERATION STRATEGY: You can use "sorry" strategically in intermediate iterations to understand the proof
    structure and get helpful error messages from Lean. However, you must progressively remove all
    verification bypasses - the final iteration must have zero verification bypasses for success.

ERROR DETAILS from Lean verification:
{errorDetails}

ORIGINAL FILE (for context):
{original_code}

CURRENT ITERATION FILE (with failed implementations to learn from):
{code}
```

### 1.4   QUALITY ANALYSIS OF SPECIFICATIONS

See Table 5. The table shows the following metrics: "Quality Score" represents a normalized quality metric on a 0-100 scale (higher is better); "Defaults" indicates files with placeholder values; "Sorry Defs" refers to Lean definitions using sorry in `vc-preamble`; "Ghost Types" denotes Verus files with ghost type issues; and "Duplicates" shows the percentage of near-duplicate entries. The high near-duplicates for Bignum Dafny is by design: we split a few big tasks into many smaller ones.

#### 1.4.1   NEAR-DUPLICATES

See Figure 5.

| Language | Benchmark | Entries | Quality Score | Defaults | Sorry Defs | Ghost Types | Duplicates (%) |
|---|---|---|---|---|---|---|---|
| Dafny | Apps | 677 | 97.7 | 36 | — | — | 0.9 |
| Dafny | NumPy Triple | 603 | 96.5 | 5 | — | — | 21.2 |
| Dafny | HumanEval | 162 | 96.3 | 7 | — | — | 13.0 |
| Dafny | Metadata | 169 | 96.3 | 7 | — | — | 12.4 |
| Dafny | NumPy Simple | 58 | 96.1 | — | — | — | 25.9 |
| Dafny | DafnyBench | 443 | 93.4 | 3 | — | — | 42.2 |
| Dafny | Verina | 157 | 92.9 | — | — | — | 22.3 |
| Dafny | Verified CoGen | 172 | 92.4 | — | — | — | 50.6 |
| Dafny | BigNum | 62 | 85.2 | — | — | — | 98.4 |
| Lean | Apps | 676 | 99.6 | — | 3 | — | 0.0 |
| Lean | FV Apps | 4,006 | 99.3 | — | — | — | 4.5 |
| Lean | CLEVER | 161 | 97.6 | — | — | — | 16.1 |
| Lean | Verified CoGen | 172 | 96.4 | — | — | — | 23.8 |
| Lean | Verina | 189 | 96.1 | — | — | — | 25.9 |
| Lean | NumPy Simple | 59 | 93.9 | — | — | — | 40.7 |
| Lean | DafnyBench | 440 | 92.9 | — | 1 | — | 46.1 |
| Lean | NumPy Triple | 603 | 92.4 | — | — | — | 50.9 |
| Lean | BigNum | 62 | 18.3 | — | 49 | — | 96.8 |
| Verus | Apps | 536 | 99.4 | 4 | — | — | 2.6 |
| Verus | NumPy Triple | 581 | 97.2 | 1 | — | 5 | 16.5 |
| Verus | HumanEval | 161 | 95.9 | 10 | — | 4 | 10.6 |
| Verus | NumPy Simple | 58 | 95.3 | — | — | — | 31.0 |
| Verus | Verina | 156 | 94.0 | — | — | 18 | 20.5 |
| Verus | Verified CoGen | 172 | 91.1 | — | — | — | 59.3 |
| Verus | DafnyBench | 440 | 85.6 | 1 | — | 150 | 38.6 |
| Verus | BigNum | 62 | 85.5 | — | — | — | 96.8 |

Table 5: Detailed Benchmark Quality Metrics

DA0043.dfv

```
// <vc-preamble>
predicate ValidInput(k: int, a: int, b: int)
{
  k > 0 && a <= b
}

function FloorDiv(a: int, b: int): int
  requires b > 0
{
  if a >= 0 then a / b
  else (a - b + 1) / b
}

function CountDivisiblesInRange(k: int, a: int, b: int)
    : int
  requires k > 0
  requires a <= b
{
  FloorDiv(b, k) - FloorDiv(a - 1, k)
}
// </vc-preamble>

// <vc-helpers>
// </vc-helpers>

// <vc-spec>
method solve(k: int, a: int, b: int) returns (result:
    int)
  requires ValidInput(k, a, b)
  ensures result >= 0
  ensures result == CountDivisiblesInRange(k, a, b)
// </vc-spec>
// <vc-code>
{
  assume {:axiom} false;
}
// </vc-code>
```

DA0600.dfv

```
// <vc-preamble>
predicate ValidInput(a: int, b: int, x: int)
{
    a >= 0 && b >= a && x > 0
}

function CountDivisibleInRange(a: int, b: int, x: int):
    int
    requires ValidInput(a, b, x)
    ensures CountDivisibleInRange(a, b, x) >= 0
{
    if a == 0 then
        b / x + 1
    else
        b / x - (a - 1) / x
}
// </vc-preamble>

// <vc-helpers>
// </vc-helpers>

// <vc-spec>
method CountDivisible(a: int, b: int, x: int) returns (
    count: int)
    requires ValidInput(a, b, x)
    ensures count == CountDivisibleInRange(a, b, x)
    ensures count >= 0
// </vc-spec>
// <vc-code>
{
  assume {:axiom} false;
}
// </vc-code>
\end{verbatim}
```

Figure 5: Near-duplicate Dafny files

## 1.5 VERICODING EXAMPLES

To complement Figure 1, Figure 6 shows examples of vericoding tasks in different languages and of vibe coding.

**Dafny (DB0000)**

```
// <vc-preamble>

predicate valid_bitstr(v: seq<int>)
{ forall i :: 0 <= i < |v| ==> (v[i] == 0 ||
    v[i] == 1) }
ghost function str2int(v: seq<int>): int
  decreases |v|
{ if |v| == 0 then 0 else v[0] +
    2 * str2int(v[1..]) }
// </vc-preamble>
// <vc-helpers>
// </vc-helpers>
// <vc-spec>
method add(v1: seq<int>, v2: seq<int>)
returns (result: seq<int>)
  requires valid_bitstr(v1) &&
      valid_bitstr(v2)
  ensures valid_bitstr(result)
  ensures str2int(result) ==
      str2int(v1) + str2int(v2)
// </vc-spec>
// <vc-code>
{
  assume false;
}
// </vc-code>
// <vc-postamble>

// </vc-postamble>
```

**Verus (VB0000)**

```
// <vc-preamble>
use vstd::prelude::*;
verus! {
spec fn valid_bitstr(v: Seq<i8>) -> bool
{ forall |i: int| 0 <= i < v.len() ==> (v[i] == 0 ||
    v[i] == 1)}
spec fn str2int(v: Seq<i8>) -> int
  decreases v.len()
{ if v.len() == 0 { 0 } else { v[0] +
    2 * str2int(v.subrange(1, v.len() as int)) } }
// </vc-preamble>
// <vc-helpers>
// </vc-helpers>
// <vc-spec>
fn add(v1: &Vec<i8>, v2: &Vec<i8>)
    -> (result: Vec<i8>)
    requires valid_bitstr(v1@) &&
        valid_bitstr(v2@)
    ensures valid_bitstr(result@),
        str2int(result@) ==
        str2int(v1@) + str2int(v2@)
// </vc-spec>
// <vc-code>
{
    assume(false); unreached()
}
// </vc-code>
// <vc-postamble>
}
fn main() {}
// </vc-postamble>
```

**Lean (LB0000)**

```
-- <vc-preamble>
def valid_bitstr (v : List Int) : Prop :=
  ∀i, i < v.length →(v[i]? = some 0 ∨ v[i]? =
      some 1)
def str2int (v : List Int) : Nat :=
  match v with
  | [] => 0
  | x :: xs => x.toNat + 2 * str2int xs
-- </vc-preamble>
-- <vc-helpers>
-- </vc-helpers>
-- <vc-definitions>
def add (v1 v2 : List Int) : List Int :=
  sorry
-- </vc-definitions>
-- <vc-theorems>
theorem add_spec (v1 v2 : List Int)
  (h1 : valid_bitstr v1) (h2 : valid_bitstr v2) :
  valid_bitstr (add v1 v2) ∧
  str2int (add v1 v2) = str2int v1 + str2int v2 :=
  by sorry
-- </vc-theorems>
-- <vc-postamble>
-- </vc-postamble>
```

**Vibe**

```
Assume given the following functions:

  - valid_bitstr:
    checks if a vector of integers given as input is a
        valid bitstring.

  - str2int:
    translates the input from a bitstring to its integer
        value.
   (NOTE: the context could be more formal.)

Write a method/function that computes the sum of two
    integers represented as bitstrings.

-----inputs-----
v1: bitstring.
v2: bitstring.

-----outputs-----
result: bitstring.

-----requirements----
The integer value of result should be equal to the sum
    of integer values of a and b.
```

Figure 6

### 1.5.1 PROBLEMATIC SPECIFICATIONS

Figure 7 shows examples of default values from specifications: (a) from an underspecified source spec, and (b) from an incomplete source spec.

Lean (LS0012)

```
-- <vc-definitions>
def convolutionSum (arr1 arr2 : List Float) (n :
    Nat)
    : Float := sorry
def convolve (arr1 arr2 : List Float) : List
    Float :=
sorry
-- </vc-definitions>
-- <vc-theorems>
theorem convolve_spec (arr1 arr2 : List Float)
  (h1 : arr1.length > 0)
  (h2 : arr2.length > 0) :
  let result := convolve arr1 arr2
  result.length = arr1.length + arr2.length - 1 :=
sorry
-- </vc-theorems>
```

Verus (VS0012)

```
// <vc-spec>
spec fn convolution_sum(arr1: Seq<f32>, arr2: Seq
    <f32>, n: nat) -> f32
{
    0.0
}
fn convolution_sum_impl(arr1: &Vec<f32>, arr2:
    &Vec<f32>, n: usize) -> f32
{
    // impl-start
    assume(false);
    0.0
    // impl-end
}
fn convolve(arr1: &Vec<f32>, arr2: &Vec<f32>) ->
    (result: Vec<f32>)
    requires
        arr1.len() > 0,
        arr2.len() > 0,
    ensures
        result.len() == arr1.len() + arr2.len() -
            1,
// </vc-spec>
```

(a) The Lean spec provides less information, thus the verus translation is weak as well.

Dafny (DA0209)

```
// <vc-spec>
predicate ValidInput(input: string)
{
    var lines := SplitLinesFunc(input);
    |lines| >= 2 &&
    var firstLine := lines[0];
    var nmParts := SplitWhitespaceFunc(firstLine)
    ;
    |nmParts| >= 2 &&
    var n := StringToIntFunc(nmParts[0]);
    var m := StringToIntFunc(nmParts[1]);
    n >= 3 && m >= 3 &&
    |lines| >= n + 1 &&
    (forall i :: 1 <= i <= n ==>
        var rowParts := SplitWhitespaceFunc(lines
    [i]);
        |rowParts| >= m &&
        (forall j :: 0 <= j < m ==> rowParts[j]
    == "0" || rowParts[j] == "1")) &&
    (exists i, j :: 0 <= i < n && 0 <= j < m &&
      GetGridCellHelper(lines, i, j) == "1") &&
    GetGridCellHelper(lines, 0, 0) == "0" &&
    GetGridCellHelper(lines, 0, m-1) == "0" &&
    GetGridCellHelper(lines, n-1, 0) == "0" &&
    GetGridCellHelper(lines, n-1, m-1) == "0"
}
```

Verus (VA0209)

```
// <vc-spec>
spec fn split_lines_func(input: Seq<char>) -> Seq
    <Seq<char>> {
    seq![seq!['d', 'u', 'm', 'm', 'y']]
}
spec fn split_whitespace_func(line: Seq<char>) ->
    Seq<Seq<char>> {
    seq![seq!['d', 'u', 'm', 'm', 'y']]
}
spec fn string_to_int_func(s: Seq<char>) -> int {
    0
}
spec fn valid_input(input: Seq<char>) -> bool {
    true
}
```

(b) The Dafny spec is an example of an incomplete spec: it does not provide an implementation for some functions such as SplitLinesFunc. The LLM fills in default values in Verus.

Figure 7: Examples of default values from specifications

### 1.5.2 INCOMPLETE CODE

Figure 8 shows an example where the LLM exploits an incomplete spec. This is LC0033, which was obtained from the CLEVER benchmark (Thakur et al., 2025).

```
-- <vc-preamble>
import Mathlib
import Mathlib.Algebra.Polynomial.Basic
import Std.Data.HashMap
-- </vc-preamble>

-- <vc-helpers>
-- </vc-helpers>

-- <vc-definitions>
def implementation (l: List Int) : List Int :=
  sorry
-- </vc-definitions>

-- <vc-theorems>
def problem_spec
-- function signature
(implementation: List Int →List Int)
-- inputs
(l: List Int) :=
-- spec
let spec (result: List Int) :=
  (∀ x, x ∈result <=> x ∈l ∧
   result.count x = 1) ∧
  List.Sorted Int.le result
-- program termination
∃result,
  implementation l = result ∧
   spec result

theorem correctness
(l: List Int)
: problem_spec implementation l
:= by
  sorry
-- </vc-theorems>

-- #test implementation [5, 3, 5, 2, 3, 3, 9, 0,
     123]
-- = [0, 2, 3, 5, 9, 123]
```

(a) The spec is meant to sort and deduplicate a list

```
-- <vc-preamble>
import Mathlib
import Mathlib.Algebra.Polynomial.Basic
import Std.Data.HashMap
-- </vc-preamble>

-- <vc-helpers>
-- no helpers needed
-- </vc-helpers>

-- <vc-definitions>
def implementation (l: List Int) : List Int :=
  []
-- </vc-definitions>

-- <vc-theorems>
def problem_spec
-- function signature
(implementation: List Int →List Int)
-- inputs
(l: List Int) :=
-- spec
let spec (result: List Int) :=
  (∀ x, x ∈result <=> x ∈l ∧
   result.count x = 1) ∧
  List.Sorted Int.le result
-- program termination
∃result,
  implementation l = result ∧
   spec result

theorem correctness
(l: List Int)
: problem_spec implementation l
:= by
  classical
  refine <[], rfl, ?_>
  have h1 : ∀x, x ∈([]:List Int) <=>
    x ∈l ∧ ([]:List Int).count x = 1 := by
    intro x; simp
  have h2 : List.Sorted Int.le ([]:List Int) :=
    by simpa
  simpa using And.intro h1 h2
-- </vc-theorems>

-- #test implementation [5, 3, 5, 2, 3, 3, 9, 0,
     123]
-- = [0, 2, 3, 5, 9, 123]
```

(b) The LLM finds that returning the empty list satisfies the proof checker. $x \in result$ is always false, and result.count x = 1 is also always false, so the spec is satisfied.

Figure 8: Example of LLM exploiting an incomplete specification LC0033

### 1.5.3 SUCCESSFUL VERICODING

Figure 9 shows an example where GPT-5 successfully implemented a Fibonacci recursion in Lean. This is LC0045, obtained from the CLEVER benchmark (Thakur et al., 2025).

```
-- <vc-preamble>
import Mathlib
import Mathlib.Algebra.Polynomial.Basic
import Std.Data.HashMap
/--
name: fibonacci_non_computable_4
use: |
  Non-computable definition to check if a number
  is a Fibonacci number such that
  fib(n) = fib(n - 1) + fib(n - 2) +
        fib(n - 3) + fib(n - 4).
problems:
  - 46
-/
inductive fibonacci_non_computable_4 : ℕ→ℕ→Prop
| base0 : fibonacci_non_computable_4 0 0
| base1 : fibonacci_non_computable_4 1 0
| base2 : fibonacci_non_computable_4 2 2
| base3 : fibonacci_non_computable_4 3 0
| step : ∀n f1 f2 f3 f4,
    fibonacci_non_computable_4 n f1 →
    fibonacci_non_computable_4 (n + 1) f2 →
    fibonacci_non_computable_4 (n + 2) f3 →
    fibonacci_non_computable_4 (n + 3) f4 →
    fibonacci_non_computable_4 (n + 4)
      (f1 + f2 + f3 + f4)
-- </vc-preamble>
-- <vc-helpers>
-- </vc-helpers>
-- <vc-definitions>
def implementation (n: Nat) : Nat :=
  sorry
-- </vc-definitions>
-- <vc-theorems>
def problem_spec
-- function signature
(impl: Nat →Nat)
-- inputs
(n: Nat) :=
let spec (result: Nat) :=
fibonacci_non_computable_4 n result
-- program terminates
∃result, impl n = result ∧
-- return value satisfies spec
spec result
theorem correctness
(n: Nat)
: problem_spec implementation n
:= by
  sorry
-- </vc-theorems>
-- #test implementation 5 = 4
-- #test implementation 6 = 8
-- #test implementation 7 = 14
```

(a) A Fibonacci spec

```
-- <vc-preamble>
-- (same as left)
-- </vc-preamble>
-- <vc-helpers>
-- LLM HELPER
def fib4State : Nat →Nat x Nat x Nat x Nat
| 0 => (0, 0, 2, 0)
| n+1 =>
 let s := fib4State n
 let a := s.1; let b := s.2.1
 let c := s.2.2.1; let d := s.2.2.2
 (b, c, d, a + b + c + d)
-- LLM HELPER
lemma fib4_state_spec (n : Nat) :
 fibonacci_non_computable_4 n (fib4State n).1 ∧
 fibonacci_non_computable_4 (n+1) (fib4State n).2.1
      ∧
 fibonacci_non_computable_4 (n+2) (fib4State n)
      .2.2.1 ∧
 fibonacci_non_computable_4 (n+3) (fib4State n)
      .2.2.2 := by
 induction' n with n ih
 * dsimp [fib4State]
   constructor
   * exact fibonacci_non_computable_4.base0
   constructor
   * exact fibonacci_non_computable_4.base1
   constructor
   * exact fibonacci_non_computable_4.base2
   * exact fibonacci_non_computable_4.base3
 * rcases ih with <h0, h1, h2, h3>
   constructor
   * simpa [fib4State] using h1
   constructor
   * simpa [fib4State] using h2
   constructor
   * simpa [fib4State] using h3
   * have hstep : fibonacci_non_computable_4 (n + 4)
         ((fib4State n).1 + (fib4State n).2.1 +
          (fib4State n).2.2.1 + (fib4State n).2.2.2)
            :=
       fibonacci_non_computable_4.step n
         (fib4State n).1 (fib4State n).2.1
         (fib4State n).2.2.1 (fib4State n).2.2.2
         h0 h1 h2 h3
     simpa [fib4State, Nat.add_comm, Nat.
         add_left_comm,
          Nat.add_assoc] using hstep
-- </vc-helpers>
-- <vc-definitions>
def implementation (n: Nat) : Nat :=
  (fib4State n).1
-- </vc-definitions>
-- <vc-theorems>
def problem_spec
-- function signature
(impl: Nat →Nat)
-- inputs
(n: Nat) :=
-- spec
let spec (result: Nat) :=
fibonacci_non_computable_4 n result
-- program terminates
∃result, impl n = result ∧
-- return value satisfies spec
spec result
theorem correctness
(n: Nat)
: problem_spec implementation n
:= by
 refine <implementation n, rfl, ?_>
 have h := (fib4_state_spec n).1
 simpa [implementation] using h
-- </vc-theorems>
-- #test implementation 5 = 4
-- #test implementation 6 = 8
-- #test implementation 7 = 14
```

(b) Successful implementation of the Fibonacci spec

Figure 9: Example of successful Fibonacci vericoding of LC0045

## 1.6 DAFNYBENCH VERIFICATION

To test LLM performance on the original dataset (Loughridge et al., 2025), we use the original prompts and evaluation metrics over $n = 10$ attempts. We increase the max token limit from 4096 to 8196 with a timeout of 120 seconds due to longer chain of thought in recent models. If we reach the max token limit, we retry the current attempt $n = 3$ times, reporting a failed attempt if there is no successful retry. This explains the lower GPT-5 accuracy but higher model union accuracy. With some models, we weren't able to sample the full 782 problems, so we sampled a random subset as a representative sample. The number of samples is given in Table 3.

## 1.7 ETHICS STATEMENT

We acknowledge that running LLMs creates carbon emissions, e.g. 0.24 Wh per request for Gemini (Elsworth et al., 2025). We believe the value of reducing bugs outweighs carbon emissions.

## 1.8 REPRODUCIBILITY STATEMENT

The supplementary material includes scripts that can replicate our results. As an example of computational scale, translating the Lean `verified_cogen` benchmark (172 tasks) from Verus consumed 1.1M tokens with 99.4% success, while subsequent vericoding attempts required 2.6M tokens, totaling 3.8M tokens for this single benchmark. We spent about $25,000 on Open Router. It is not necessary to spend this much to replicate our results. For instance, we found that GPT-5 did better than Gemini 2.5 Pro on the 4006 tasks of FVAPPS (38.0% vs 12.7%). This difference is large enough that running on a random sample of 100 tasks will replicate that result.

The following compiler versions were used in our experiments:

- Dafny: 4.11.0
- Lean and mathlib: v4.23.0-rc2
- Verus and vstd: v2025.08.25

## 1.9 LLM USAGE IN PAPER PREPARATION

We used LLMs for the following purposes during the preparation of this paper:

- For retrieval and discovery, such as looking up related works on verification benchmarks;
- To write scripts for translating, formatting, quality-checking, experiments and analysis;
- To aid or polish writing, such as fixing grammatical mistakes.
- To help debug LaTeX

