# OpenReview forum: "A Benchmark for Vericoding: Formally Verified Program Synthesis"
_ICLR.cc/2026/Conference — Submitted to ICLR 2026_

### Official Review · Reviewer_JR7g · 2025-10-19

**Soundness:** 3
**Presentation:** 2
**Contribution:** 1
**Rating:** 2
**Confidence:** 4

**Summary:**

The paper's main contribution is to introduce a benchmark for "vericoding" (defined in this paper as generating code + proof given a formal spec). Data examples in this benchmark were curated by pulling together existing vericoding benchmarks in different formal languages (Lean, Dafny, Verus) and using LLMs to translate between them, as well as autoformalizing coding benchmarks such as APPS. The authors also evaluated many off-the-shelf LLMs such as GPT-5 on the new benchmark and performed analysis on the results.

**Strengths:**

I agree with the high-level motivation of this work. Generating formally verifiable code is important and will be more important in the near future as LLMs become better at code generation. Well-constructed benchmarks in this space will unleash research progress.

**Weaknesses:**

However, I'm not convinced by the quality of the proposed benchmark and the method for constructing it. Specifically:

* Most examples in the benchmark were either directly from existing benchmarks in different languages (Dafny, Lean, Verus) or obtained by using LLMs to translate between them. It is not clear what value the new benchmark brings or why would a subsequent work evaluate on this new benchmark instead of on multiple existing benchmarks individually.

* The quality of the benchmark is not clear. Only a subset of the translated problems are manually inspected by humans. Many translated specs may be too weak (Line 370). The authors argue that perfect translation is not required; as long as a spec is syntactically valid, it can potentially be a useful task for training or evaluation. I agree the translation doesn't need to be perfect, but it needs to above a certain level for the benchmark to be useful.

* The new benchmark's data sources (CLEVER, Verina, HumanEval, etc.). For example, CLEVER translated HumanEval into Lean. It's not clear why the authors attempted to translate HumanEval again.

* This paper uses "Vibe coding" to refer to the dominant approach to AI code generation that does not involve formal methods. For example, in Line 24, the authors claim that Google is using "vibe coding" to generate over 30% of its software, which sounds astonishing. However, what the cited report actually says is over 30% of the code was generated by AI and accepted by human engineers. The research on AI for code generation has a long history and shouldn't be referred to as "vibe coding" (a term coined by Andrej Karpathy in a tweet in 2025 and is often used in the context of LLM coding agents such as Claude Code) in a serious academic publication.

 * The paper introduced a new term "vericoding" to refer to generating code and proof given a formal spec. This task is usually called verified (or verifiable) code generation in recent papers (CLEVER, FVAPPS, Verina, etc.). Before that, program synthesis from formal specification is a well-established task in the PL community.

**Questions:**

* Why is FVAPPS categorized as a "vibe coding" benchmark instead of vericoding?

---

> ### Author Response · Authors · 2025-11-21
>
> We thank the reviewer and respond below to the weak points and questions raised.
>
> - On the value of collating tasks: We think there is value in collating all the benchmarks and standardizing them. Our benchmark and cheating detectors could be plugged into standard AI framework like inspect-ai making it easier to evaluate their performance.
>
> - On benchmark quality: To address quality concerns, we consider adding metrics like "in dataset X in language Y, a% of specs are correct, b% are too weak, c% are impossible, and d% are wrong in some other way".  These could be dynamic across iterations of the benchmark.
>
> - On Human Eval translations: Our main HumanEval source for Lean is indeed Verina. We have translated some Human Eval specs into Verus or Dafny when those were missing from original sources.
>
> - On vibe coding, vericoding and verifiable code generation: We will clarify terminology and discuss related work more thoroughly. Our paper calls AI-assisted programming "vibe coding" if it is based on informal specifications.
>
> Q: Why is FVAPPS categorized as a "vibe coding" benchmark instead of vericoding?
>
> R: We agree the distinction is not clearcut. We have categorised FVAPPS as vibe coding because we have considered that it has informal specifications as its main input. We will review our classification and make the criteria more clear in the revised version of the paper.

---

### Official Review · Reviewer_vsVC · 2025-11-01

**Soundness:** 3
**Presentation:** 3
**Contribution:** 3
**Rating:** 6
**Confidence:** 4

**Summary:**

This paper introduces a benchmark set for vericoding, which is the task of synthesizing formally verified code (including proofs) from formal specifications. The benchmark set contains benchmarks for Dafny, Verus, and Lean. The benchmarks are generated from multiple sources:
1. cross translation into target languages from existing verification benchmarks;
2. auto-formalizing vibe-coding benchmarks;
3. auto-formalizing documentation of libraries such as NumPy and BigNum.

The paper describes several ways employed to ensure the quality of the benchmarks. The paper also includes a thorough evaluation of a range of LLMs on this benchmark.

**Strengths:**

- A comprehensive vericoding benchmark set is good to have, given that there has been significant research interest in this area.
- The paper uses iterative refinement coupled with verification to improve the success rate of the translation.
- The paper identifies several additional potential issues that may hamper the quality of the benchmark, and describes ways to mitigate them.
- The experimental evaluation is quite thorough and gives a nice overview of the current landscape of LLM capability in vericoding.

**Weaknesses:**

- The language-to-language generation process is LLM-based, which is a little sloppy and itself error-prone. I understand that translating proofs between theorem provers can be challenging (and, anyway, verification can ensure the soundness of the proof), but I wonder whether translating the specifications can be done in a more rigorous manner, using analytical approaches.
- Although the paper seems to have spent non-trivial effort to ensure the quality of the benchmarks, the quality of the benchmarks is still not close to being perfect (9% weak specs and 15% incorrect translation).

**Questions:**

Could you discuss the challenge of translating specifications between theorem provers in a formal manner?

---

> ### Author Response · Authors · 2025-11-21
>
> We thank the reviewer and respond below to the weak points and questions raised.
>
> (A) Concerning the weak points.
>
> - More rigorous translations: We plan to guardrail AI translations with analytical methods in future work. Purely analytical translators between spec are also an option we consider.
>
> - Weak specs: We think our benchmark can serve as a basis from which specs can be solved, varied and improved.
>
> (B) Concerning questions:
>
> Q: Could you discuss the challenge of translating specifications between theorem provers in a formal manner?
>
> R: Thank you for the suggestion, we will discuss this in a revised version of the paper.

---

### Official Review · Reviewer_4LEx · 2025-11-02

**Soundness:** 2
**Presentation:** 2
**Contribution:** 2
**Rating:** 4
**Confidence:** 4

**Summary:**

This paper introduces a substantial vericoding benchmark, encompassing 12,504 formal specifications. These specifications are divided among Dafny (3,029), Verus (2,334), and Lean (7,141), with 6,174 being newly developed problems. Vericoding distinguishes itself from end-to-end generation by requiring Large Language Models (LLMs) to generate both implementations and their corresponding proofs from existing formal specifications, rather than inferring the specifications themselves. Current frontier LLMs demonstrate varying success rates: 82% for Dafny, 44% for Verus, and 27% for Lean.

**Strengths:**

- Unprecedented Scale: With 12,504 tasks, this benchmark is two orders of magnitude larger than existing verification benchmarks, addressing a significant gap in the field.
- Comprehensive Language Coverage: It includes both Automated Theorem Provers (ATPs) like Dafny and Verus, and Interactive Theorem Provers (ITPs) such as Lean, thereby enabling cross-paradigm comparisons.

**Weaknesses:**

- Duplication Concerns: Figure 5 highlights the presence of near-duplicate tasks, which are essentially the same problem with minor variations. This issue artificially inflates the benchmark's size and reduces its overall diversity.
- Limited Quality Validation: The study only examines 5 tasks per model and source, indicating a lack of systematic evaluation regarding the quality of the benchmark tasks.
- Insufficient Evaluation and Analysis: Figure 4's analysis is restricted to length metrics, which are obvious predictors. The paper lacks investigation into crucial aspects such as: factors contributing to task difficulty (e.g., proof complexity, theorem depth), reasons behind different models' varying performance across languages.

**Questions:**

- Is there a plan to release a cleaned version of the benchmark that addresses the 24% of problematic specifications?
- How is the inclusion of 98.4% near-duplicate tasks justified, and shouldn't these be deduplicated?
- Is every task in the benchmark provable, meaning each has a ground-truth code and proof?
- Can the manual inspection process be expanded beyond the current 5 samples, which represent less than 1% coverage?

---

> ### Author Response · Authors · 2025-11-21
>
> We thank the reviewer and respond below to the weak points and questions raised.
>
> (A) Concerning the weak points
>
> - Duplication Concerns: We acknowledge that the filtering out of duplicates deserves more attention. In some cases, for example Bignum, we have deliberately created duplicate specifications for the same functions, where different helpers are included in the context. This is intended to model real world programming experience where the developer may need to implement new functions from varying premises.
>
> - Limited Quality Validation: We plan to perform more systematic validation in future work.
>
> - Insufficient Analysis: We plan to perform more systematic difficulty analysis in future work.
>
> (B) Concerning questions:
>
> Q: Is there a plan to release a cleaned version of the benchmark that addresses the 24% of problematic specifications?
>
> R: Yes, we plan to have iterations and variations of the benchmark.
>
> Q: How is the inclusion of 98.4% near-duplicate tasks justified, and shouldn't these be deduplicated?
>
> R: We think keeping near duplicate specs in the benchmark can be valuable in some circumstances. In this particular case (BigNum), we have deliberately created similar specs to enlarge the dataset with tasks with varying context. In most sub-benchmarks the number of near-duplicates is lower. For some of our sources, e.g. Dafny Bench, we did remove many duplicates that we deemed unhelpful, e.g. various implementations of sorting methods of Fibonacci numbers.
>
> Q: Is every task in the benchmark provable, meaning each has a ground-truth code and proof?
>
> R: We don’t know. It is an interesting question to investigate in future work.
>
> Q: Can the manual inspection process be expanded beyond the current 5 samples, which represent less than 1% coverage?
>
> R: We plan more systematic procedures in future work.

---

### Official Review · Reviewer_dMe1 · 2025-11-04

**Soundness:** 1
**Presentation:** 3
**Contribution:** 2
**Rating:** 2
**Confidence:** 4

**Summary:**

This paper introduces a comprehensive and large benchmark for the automatic generation of formally verified programs given formal specifications. The benchmark spans three verification frameworks (Dafny, Verus, and Lean), covering both automated theorem provers (ATPs) and interactive theorem provers (ITPs). It aggregates and translates tasks from multiple prior sources (e.g., DafnyBench, Verina, FVAPPS, HumanEval) into a unified suite of 12,504 tasks, of which 6,174 are new. The authors evaluate a broad range of LLMs (GPT-5, Claude, Gemini, etc.) on this benchmark, achieving best results in Dafny (82.2% success) and lower rates for Verus (44.3%) and Lean (26.8%). The study aims to provide a foundation for measuring LLM progress in generating provably correct code.

**Strengths:**

1. Introduces a large-scale dataset on a timely and impactful topic—AI-assisted formal verification.
2. Covers diverse verification frameworks (Dafny, Verus, Lean), enabling cross-system comparisons between ATP and ITP paradigms.
3. Provides comprehensive LLM evaluations, revealing quantitative differences across languages and models.
4. Transparent methodology with anti-cheating validation and manual inspection reports.
5. Clear writing.

**Weaknesses:**

1. Conceptual limitation of the “vericoding” definition: The paper defines vericoding narrowly as code and proof generation from formal specifications, explicitly excluding specification generation. However, specification writing is the main bottleneck in formal verification and a key component of prior benchmarks such as CLEVER and VERINA. By ignoring spec generation, this benchmark captures only a partial and arguably easier slice of the full verified-coding problem. This weakens the practical and conceptual impact of the work.

2. I find the benchmark quality to be weak, for reasons detailed below. This is a major limitation, especially because the work targets formal verification—a domain that demands the highest level of rigor. If the benchmark itself includes incomplete or incorrect specifications, it cannot faithfully measure models’ verification capabilities and may even lead to misleading or unsafe outcomes if such models are later used in production settings.
    - The spec translation and validation rely on an LLM judge to assess semantic equivalence, which may not guarantee correctness. The verifier only ensures syntactic validity.
    - I appreciate the authors’ manual inspection and being upfront about weak specification or other problems with the benchmark. However, the description reads vague to me: “Verus showed *many* weak specs due to spec translation issues, e.g. deviations from the original BigNum specs in Dafny. Lean displayed *occasional* redundant lemma additions during vericoding, and had *some* weak specs from the original sources.”. Could you provide quantitative measures?
    - The authors also did not seem to talk about how specifications identified as low quality are processed. Are they still included in the benchmark, or are they fixed? My impression is that they are still included.
    - The current quality score of the benchmark is a weighted average of several defined criterias, which only consider obvious cases (including defaults, sorry defs, ghost types, duplication), but do not measure the semantic quality of the specification.

3. The evaluation settings are unclear and lack important baselines.
    - The paper does not specify how many refinement iterations are allowed, what error messages are fed back, or whether these differ across verification frameworks. Moreover, how do these choices impact performance?
    - Only general-purpose LLMs are tested. No experiments with proof-specialized approaches (e.g., DeepSeek-Prover-V2 [1], Goedel-Prover-V2 [2], SAFE [3], AutoVerus [4], AlphaVerus [5]) are included.

4. Overall, the benchmark emphasizes scale, but size alone does not determine effectiveness. Beyond quality, a benchmark should be diverse enough to cover a broad spectrum of difficulty levels to meaningfully differentiate model capabilities. In fact, without such diversity, a large benchmark can even become a costly burden for evaluation, obscuring rather than clarifying model progress. In this regard, I think the paper is doing a good job in including both ATPs and ITPS. However, In Table 2, although multiple benchmark sources are combined, the paper does not discuss the degree of overlap among these sources or the relative difficulty of the tasks they contribute. It is therefore unclear whether the enlarged dataset truly expands task diversity or merely aggregates similar problems, which raises questions about how well the benchmark can measure real advances in verification performance. For example, in Table 3, the performance trend on the Verina subset closely mirrors the overall average, suggesting limited incremental value. If that is the case, why not evaluate directly on the Verina subset instead?

[1] Ren, Z. Z., et al. "Deepseek-prover-v2: Advancing formal mathematical reasoning via reinforcement learning for subgoal decomposition." arXiv preprint arXiv:2504.21801 (2025).

[2] Lin, Yong, et al. "Goedel-prover-v2: Scaling formal theorem proving with scaffolded data synthesis and self-correction." arXiv preprint arXiv:2508.03613 (2025).

[3] Chen, Tianyu, et al. "Automated proof generation for rust code via self-evolution." ICLR 2025.

[4] Yang, Chenyuan, et al. "Autoverus: Automated proof generation for rust code." Proceedings of the ACM on Programming Languages 9.OOPSLA2 (2025): 3454-3482.

[5] Aggarwal, Pranjal, Bryan Parno, and Sean Welleck. "Alphaverus: Bootstrapping formally verified code generation through self-improving translation and treefinement." ICML 2025.

**Questions:**

Please address my comments in the “Weakness” section.

---

> ### Author Response · Authors · 2025-11-21
>
> We thank the reviewer and respond below to the weak points raised.
>
> 1. Vericoding limitation:
> We think vericoding as defined in the paper poses its specific challenges that deserve to be studied independently of other parts of the formally verified code generation pipeline.
>
> 2. Benchmark quality:
> We agree our quality assurance process needs to be improved in future work. Our focus in this paper is more on gathering many vericoding challenges that are sound from a logical point of view. As shown by our results, they are still meaningful challenges for the LLM. We think the benchmark could also be useful as a training set.
>
> 3. Evaluation setting:
> We will make the experimental setting more clear in a revised version of the paper. In ongoing work, we are also experimenting with specialised AI tools, but we think it is valuable to test the benchmark on a selection of general purpose models.
>
> 4. Near-duplicates issue:
> We did not run cross-source diversity checks, as we expect for example the Verina bench to be sufficiently different from the Verified Cogen bench. Also, it can be argued that two closely resembling specifications may pose significantly distinct vericoding challenges.

---

### Meta-Review · Area_Chair_3edb · 2026-01-02

**Summary:**

The paper presents a new benchmark for code generation with corresponding formal proofs by combining and translating several existing datasets. While the topic is timely and potentially impactful, all reviewers have similar concerns about the benchmark quality.

Reviewers (dMe1, vsVC) note that the LLM-based translation between verification frameworks does not guarantee correctness, and the paper provides no quantitative measures to assess benchmark quality (dMe1, 4LEx). Multiple reviewers(vsVC, JR7g) also report that manual inspection reveals many weak specifications, and the quality is still not close to being perfect. Additional concerns include duplicate samples in the benchmark(dMe1, 4LEx), limited evaluation and analysis(dMe1, 4LEx)

The authors’ rebuttal provides some clarifications, claiming that duplicate samples can increase difficulty and will improve benchmark quality as well as evaluation settings in future works. However, no new evidence or data analysis is presented that would change the reviewers’ assessments.

Given the consistent concerns across all reviewers and the limited impact of the rebuttal, the appropriate decision is reject.

**Reviewer Concerns:**

The concerns about the usage of terms "Vibe coding" and “vericoding” might be addressed as the author will clarify terminology and discuss related work more thoroughly.

Concerns about the benchmark quality, evaluation, and duplicate samples might not have been addressed, as the rebuttal does not provide new quantitative measures or evaluation results.

**Reviewer Scores:**

Since the rebuttal does not provide new evidence to address the shared concerns about benchmark quality, all reviewers are unlikely to change their scores.

---

### Decision · Program_Chairs · 2026-01-26

Reject